# Exploring the Chemical Space of Kawakawa Leaf (*Piper excelsum*)

**DOI:** 10.3390/nu14235168

**Published:** 2022-12-05

**Authors:** Ramya Jayaprakash, Farha Ramzan, Jennifer L. Miles-Chan, Meika Foster, Richard F. Mithen, Chris Pook

**Affiliations:** 1Liggins Institute, Waipapa Taumata Rau—The University of Auckland, 85 Park Road, Private Bag 92019, Auckland 1142, New Zealand; 2Human Nutrition Unit, Waipapa Taumata Rau, The University of Auckland, 85 Park Road, Private Bag 92019, Auckland 1142, New Zealand; 3Edible Research Ltd., Ohoko 7475, New Zealand; 4AuOra Ltd., Wakatū Incorporation, Nelson 7010, New Zealand

**Keywords:** metabolomics, molecular networking, LC-MS/MS, phytochemicals, toxicity, functional food, natural products, ethnobotany, ethnopharmacology

## Abstract

The chemical profiles of kawakawa (*Piper excelsum*) leaves were analysed through targeted and non-targeted LC-MS/MS. The phytochemical profile was obtained for both aqueous extracts representative of kawakawa tea and methanolic extracts. Sixty-four compounds were identified from eight leaf sources including phenylpropanoids, lignans, flavonoids, alkaloids and amides. Eight of these compounds were absolutely quantified. The chemical content varied significantly by leaf source, with two commercially available sources of dried kawakawa leaves being relatively high in phenylpropanoids and flavonoids compared with field-collected fresh samples that were richer in amides, alkaloids and lignans. The concentrations of pharmacologically active metabolites ingested from the traditional consumption of kawakawa leaf as an aqueous infusion, or from novel use as a seasoning, are well below documented toxicity thresholds.

## 1. Introduction

Kawakawa (*Piper excelsum* G. Forst.) is a shrub endemic to Aotearoa New Zealand and is a taonga, or treasure, of great cultural importance to Māori. Kawakawa leaves are used in Rongoā Māori (Māori traditional medicine) to treat a wide range of ailments such as genitourinary infections, gastrointestinal discomforts, respiratory disorders, toothache and skin diseases [1]. Dried leaves for use as an herbal tea or seasoning are commercially available, and there is interest in exploring the use of kawakawa for therapeutic preparations and as a functional food and beverage ingredient. Solvent extracts of kawakawa leaves have shown anti-inflammatory potential through the reduction of IL-6 and NF-κB gene expressions in human cell lines [2]. Similarly, kawakawa amides influenced glucose and fatty acid uptake by human epithelial Caco-2 cells [3]. Our group recently reported findings from the first clinical kawakawa intervention study, which indicated that kawakawa tea consumption may modulate postprandial glucose metabolism [3]. It is of interest to investigate the kawakawa chemical space to identify pharmacologically active compounds that may be responsible for this and other potentially beneficial effects.

Two previous studies have reported the chemical composition of dried and fresh kawakawa leaves. Obst et al. [4] identified 24 features with the NMR and LC-MS/MS analysis of kawakawa ethanolic extracts including amides, flavonoid glucosides, phenylpropanoids and lignans. Butts et al. [5] reported 21 features in the LC-MS and NMR analysis of aqueous and ethanolic extracts of both commercially available dried leaves and fresh leaves from the East Cape regions of Te Ika-a-Māui, the North Island of New Zealand. Neither study reported absolute quantitative results. The pharmacologically active compounds putatively identified include the amide piperine and several of its analogues, which demonstrated a range of effects on human health [6]. These include enhancing the bioavailability of other pharmacologically active compounds, such as curcumin [7], and the activation of the transient receptor potential cation channel vanilloid subfamily member 1 [TRPV1] receptor [8,9]. Other putative kawakawa compounds include the phenylpropanoids myristicin and elemicin. These compounds are abundant in many other foods, but particularly in nutmeg. Cases of nutmeg abuse have been reported by people in pursuit of psychotropic effects alleged to result from the metabolism of the phenylpropanoids to amphetamine analogues [10]. However, forensic analyses of urine from an abuse case and from rodent models of phenylpropanoid metabolism do not support this theory [11]. There is no evidence of kawakawa toxicity from its traditional use by Māori [1].

In this study, we report a comprehensive analysis of the chemical compounds extracted from both commercially available dried kawakawa leaves and from fresh leaves sourced from different geographic locations in Aotearoa New Zealand at different times of the year. Mass spectrometry-based molecular networking was employed to visualise the complexity of the kawakawa leaf chemistry, and several of the most abundant compounds were quantified to inform regulatory approval for extending the use of kawakawa leaf as a food product. We provide details of 58 compounds in leaves of *P. excelsum* that have not been previously reported, the absolute quantification of a subset of these compounds and preliminary data on the importance of seasonal and geographic variation in determining the chemical profile of kawakawa.

## 2. Materials and Methods

### 2.1. Solvents and Chemicals

Authentic standards were obtained as follows: myristicin, pellitorine and fagaramide (Sigma-Aldrich, Sydney, Australia); elemicin, apiole, piperine and vitexin (Bio-strategy, Auckland, New Zealand); yangambin (ALB technology, Kowloon, Hong Kong); piperlongumine (AK Scientific Union City, CA, USA) and podophyllotoxin (AFT Pharmaceuticals, Auckland, New Zealand). Acetonitrile (Optima LC-MS grade), formic acid, and ammonium formate (Thermo Fisher Scientific, Auckland, New Zealand). Type 1 water was generated from a Millipore unit (Merck Millipore, Auckland, New Zealand).

### 2.2. Preparation of Standards

A mix containing myristicin (1040 μM), yangambin (1120 μM), elemicin (48 μM), vitexin (23 μM), pellitorine (45 μM), fagaramide (40 μM), dopamine (65 μM), trigonelline (73 μM) and the podophyllotoxin internal standard [IS] (12 μM) was prepared and serially diluted. Accurate mass-to-charge ratio (*m/z*), retention time and MS/MS spectra were added to an in-house MS/MS library based on the 2020 NIST library (National Institute of Standards & Technology, Gaithersburg, MD, USA).

### 2.3. Kawakawa Leaf Collection

Fresh kawakawa leaves were harvested by Wakatū Incorporation in 2020 and 2021 from sites in Te Tauihu, the Northern region of Te Wai Pounamu (45.1° S, 169.8° E), the South Island of Aotearoa New Zealand (Table 1). Samples from Mohua, Golden Bay, were collected in October 2020 from Whangamoa in early and late January, early and late March and April 2021, as well as from Arapaoa Island (Lat: 41.1° S; Lon: 174.3° E) and Pōhara (Lat: 40.8° S; Lon: 172.8° E) in late 2021. These sites were chosen as representative sites within Te Tauihu. Fresh leaves were washed in cold water, frozen at −20 °C, lyophilised and ground to powder. Commercially available dried kawakawa leaves harvested from unspecified locations in Te Ika A Māui (39.2° S, 175.6° E), Aotearoa New Zealand’s North Island, were acquired in 2020 from ŌKU NZ Native Herbal Products Ltd. (Hamilton, New Zealand) and Phytomed Medicinal Herbs Ltd. (Auckland, New Zealand).

### 2.4. Extraction and Sample Preparation for LC-MS/MS

#### 2.4.1. Aqueous Extraction

Lyophilised leaf samples were finely ground in liquid nitrogen. A 10 mg mass of leaf powder was weighed into a 10 mL amber glass headspace vial, a 10 mL volume of preheated Type 1 water containing IS (12 μM) was added and the vial was capped and maintained at 80 °C for 10 min. Vials were allowed to cool to room temperature, and the particulate material pelleted by centrifugation at 10,000× *g* for 5 min at 4 °C. A volume of supernatant was transferred to a polypropylene autosampler vial and capped for LC-MS analysis. All extractions and analyses were conducted in triplicates.

#### 2.4.2. Solvent Extraction

Ultrasound-assisted methanol extraction was carried out in triplicate in 4 mL amber glass vials. Masses of 100 mg of the ŌKU commercial sample, and the Arapaoa and Pōhara fresh samples were ground as above and extracted with 2 mL of 80% MeOH ultrasonication for 30 min in an ultrasonic bath (Soniclean 80T, Soniclean Pty Ltd., Adelaide, Australia) on the highest power setting. The extract was centrifuged at 5000× *g* for 5 min, and 1.5 mL of supernatant was collected. Another 1.5 mL of 80% methanol was added, and the pellet was re-extracted and centrifuged. This re-extraction was repeated until three volumes of supernatant were combined (4.5 mL). Extracts were dried in a SpeedVac centrifugal concentrator (Savant SC250EXP; Thermo Scientific, Hampton, VA, USA) coupled with refrigerated vapour trap (Savant RVT5105, Thermo Scientific, Hampton, VA, USA) and the dry residue massed and resuspended in 10 mL 80% MeOH. A fourth extraction volume was obtained and analysed directly to assess extraction completeness.

### 2.5. LC-MS/MS Instrument and Conditions

The liquid chromatograph comprised an Accela 1250 pump (Thermo Fisher Scientific, Waltham, MA, USA), a Thermo TriPlus autosampler (Thermo Fisher Scientific, Hampton, VA, USA), a HotDog5090 column oven and a Phenomenex Kinetex biphenyl analytical column (150 mm × 2.1 mm × 1.7 μm) held at 30 °C. Flow was 0.3 mL/min and the injection volume was 5 µL. Mobile phase was MilliQ with 5 mM ammonium formate, 0.1% formic acid (A) and 100% acetonitrile with 0.1% formic acid (B). The mass spectrometer was a Q-Exactive (Thermo Scientific, Dreieich, Germany) with a heated electrospray ionisation [ESI] source. Source voltage was 3.5 kV, S-lens RF level 50, heated capillary temperature 263 °C, AGC target 1e6 and resolution 17,500. Mass spectra were acquired in positive polarity and centroid mode.

Data for qualitative analysis were acquired in data-dependent acquisition mode [DDA]. The gradient was 0–100% B in 14 min, held for 1 min and then re-equilibrated to 0% B from 15 to 20 min. Scan range was *m/z* 50–700 with normalised collision energy (NCE) of 30 eV. Chromatographically resolved features with MS/MS data from the aqueous extracts were annotated using a combination of chemoinformatics and manual inspection. Features were annotated using spectral matching in MS-DIAL v4.7 and an in-house MS/MS library based on the 2020 NIST library [12]. A molecular network was exported from MS-DIAL with a mass tolerance of 0.025 Da and a 75% similarity threshold. Features were also manually annotated by comparison with the existing literature on the chemical space of the Piper genus. Adducts and other duplicates were filtered out, as were features not directly connected to an annotated node. The MS/MS spectra were loaded into MS-FINDER v3.52 for computational formula predictions, fragment annotations and structural elucidation [13,14]. Final identities were assigned based on the above information, together with manual inspection of the MS/MS spectra to identify fragments and structural moieties common to multiple features and classes of compound. Cytoscape v3.9.0 was used to visualise the final network [15].

For comparative analysis of aqueous leaf extracts, spectral data were collected using all-ion fragmentation [AIF]. The full scan range was from *m/z* 75 to 850, followed by two AIF scans from *m/z* 56.7 to 750 with 10 and 35 eV NCE, respectively. The gradient started at 0% B, ramped up to 95% B at 16 min, held at 95% B until 18 min, then re-equilibrated at 0% B from 20 to 25.5 min.

MS conditions for quantitative analysis were the same as for qualitative analysis. The gradient was 0% B for 1 min, then 0–95% B at 13 min, held for 1 min, then re-equilibrated at 0% B from 14 to 20 min. The same method was used for both methanolic and aqueous extracts.

### 2.6. Data Processing, Visualisation and Statistical Analysis

Data were collected and processed using Thermo Xcalibur software (Version 4.2.47). MetaboAnalyst (Version 5.0) was used to create clustergrams to visualise the variations in metabolite abundance between different sample types [16]. Absolute quantitative data were analysed using one-way ANOVA followed by a two-stage step-up method of Benjamini, Krieger and Yekutieli for multiple pairwise comparisons [17] using GraphPad Prism Version 9.1.2 (GraphPad, San Diego, CA, USA).

## 3. Results

### 3.1. Qualitative Metabolic Profiling of Kawakawa Aqueous Extracts

Using MS/MS spectra from eleven authentic standards, we identified trigonelline **2**, dopamine **3**, vitexin **7**, fagaramide **24**, yangambin **32**, elemicin **38**, piperine **41**, myristicin **45** and pellitorine **49** to Level 1 of the Metabolomics Standards Initiative [MSI] guidelines [18]. A further 262 features were annotated by spectral matching to reference spectra using MS-DIAL with scores >0.85. Compounds previously identified in *P. excelsum* and other *Piper* species were well represented, including phenylpropanoids, alkaloids, amides, flavonoids and lignans [4,5,19,20,21]. Examples include dihydropiperlonguminine **34**, myristicin **45**, vitexin **7**, vitexin-O-glucoside **4**, cinnamoyl piperidine **26** and cinnamoyl piperideine **16** [5], dimethoxycinnamoyl piperidine **22** [22] and piperanine **42** [19]. Features annotated as apiole and piperlongumine did not match the retention time and mass spectra of those standards.

Molecular networking was used to filter the chemical space by reducing the 1453 MS/MS spectra of chromatographically resolved features to a size that could be manually scrutinised. Nodes not directly linked to one already annotated were excluded. Adducts and other artifacts were manually filtered, and 154 features were processed using MS-FINDER to generate orthogonal annotations. Identities were assigned to a total of 66 features, including isomers.

Scrutiny of the MS/MS spectra of amides revealed patterns of fragmentation corresponding the moieties either side of the amide nitrogen. For example, the methylenedioxycinnamoyl fragment of fagaramide yields an intense ion at *m/z* 175.0395, whilst the isobutyl fragment is *m/z* 57.0699 (Appendix A). The same pattern of fragmentation was observed with piperine and piperlongumine. This pattern was combined with the information from spectral matching and in silico structural elucidation using MS-FINDER to assign amide identities to 46 features in the data (Table 2). The amides could be divided into groups based on their structure, being based on either cinnamoyl, methylenedioxycinnamoyl or fatty acid moieties. These were combined with a variety of amines such as piperidine, piperideine, pyrrolidine and isobutylamine. The cinnamoyl ring was sometimes modified with hydroxyl and methoxy groups, and cyclic amines on cinnamoyl amides sometimes exhibited hydroxyl or ketone groups. The hydrocarbon chain that links the phenyl ring of the cinnamoyl moiety with the amide varied in length and saturation.

Chemoinformatics allowed us to annotate several other features such as the catecholamine neurotransmitter dopamine, several flavonol glucosides and two lignans besides the yangambin isomers (Table 2). We often observed chromatographically resolved peaks with very similar MS1 and MS2 spectra. For example, our authentic standard for trans-fagaramide **24** coeluted and spectrally matched with the feature at 8.40 min so the one at 8.15, which was also a close spectral match, was assumed to be cis-fagaramide.

Yangambin **32**, was only observed in ACO extracts, at low abundance. Two other features which appeared a few seconds later in all other kawakawa extracts with similar MS2 spectra and much greater abundance were annotated as epiyangambin by spectral matching. The [M + NH_4_]^+^ adduct for these features was twice that of the [M + H]^+^ adduct, whereas for yangambin it was <10% (Appendix A). No reference spectra could be found for the third yangambin isomer, diayangambin, and an authentic standard was not available to purchase. Diayangambin was identified in kawakawa by Russell and Fenemore [5], so we assumed the first and most abundant peak of these two was that and the last one was epiyangambin.

Multiple features were annotated as dimethoxycinnamoyl piperidine **22**, cinnamoyl piperidine **26** and piperlonguminine **28**, among others (Table 2). To visualise the kawakawa chemical space, the annotated features are shown as a molecular network in Figure 1. Features with *m/z* 150.0553 and *m/z* 279.1222 were observed to be abundant in the kawakawa extracts but no identity could be robustly assigned to them. For clarity, only features that could be annotated with confidence are visualised.

Alkaloids and amides comprised the largest cluster in the network (Figure 1A). Cinnamoyl or methylenedioxycinnamoyl moieties combined with piperidine were the most commonly observed and connected structures in the network and occupy the centre. A subcluster can be seen at the top of the network, populated with compounds whose amide groups carry hydroxyl or ketone groups or have unsaturated bonds. Molecules on the mid-to-lower-left mostly contain methylenedioxybenzole rings with longer hydrocarbon backbones than those on the right. Fatty acid amides mostly are localised to the right edge of the network. Compounds in the middle of the network that are based on a cinnamoyl moiety are methoxylated to varying degrees.

Flavonoids comprised the second-largest cluster (Figure 1B), including vitexin and its glucosides. Dopamine **3** was clustered together with another catecholamine, salsolinol **1**, and methyldopamine **10**. Methyldopamine was also connected to the aromatic amino acid tryptophan **15**. The phenylpropanoids myristicin **44** and elemicin **37**, the lignan magnolin **33**, three amides, and the alkaloid trigonelline **2** did not cluster with any other components.

### 3.2. Differences in Kawakawa Chemistry

The differences between the chemical profiles of the eight aqueous kawakawa extracts were visualised using clustermaps for unsupervised hierarchical clustering analysis (Figure 2). Features in the extracts from commercial leaf sources, ACO and ACP, clustered together in the clustermap, indicating similarities in their phytochemical profiles, particularly in their phenylpropanoid content. Features in fresh sample extracts also clustered together and were observed to be particularly rich in amides, except for AFT4. The AFT3, 4 and 5 extracts were relatively rich in lignans. AFM extracts seem to be exceptionally rich in both phenylpropanoids and alkaloids. This latter sample is unique both in the geographic location (Mohūa, Golden Bay) and season (late October, i.e., springtime versus summer/autumn) in which it was collected.

### 3.3. The Comparison of Aqueous and Methanolic Kawakawa Extracts

Figure 3 shows the results from triplicate methanolic extractions of kawakawa. Analysis of a fourth extract showed that extraction efficiency was >99% (data not shown). Comparison of quantitative data for the aqueous and methanolic extracts of the commercial ACO kawakawa show that methanolic extraction was more effective for most compounds, as would be expected. The exceptions were myristicin and elemicin, which were slightly elevated in aqueous extracts. This is not unexpected as these two compounds can be considered to be of mid-polarity.

### 3.4. Absolute Quantitative Analysis of Aqueous and Methanolic Kawakawa Extracts

We used LC-MS to absolutely quantify myristicin, elemicin, fagaramide, diayangambin, vitexin and pellitorine in aqueous extracts of the ACO and AFT1 kawakawa samples (Figure 4A). Compared with AFT1 samples, ACO samples had significantly higher levels of myristicin (*p* < 0.01) and elemicin (*p* < 0.001). Mean (±SD) myristicin concentration in ACO was 5.28 ±1.2 μmol per gram of dry weight leaf powder [μmol g^−1^] compared with 0.88 ± 0.1 µmol g^−1^ in AFT1. Elemicin concentration in the commercial sample was 9.34 ± 0.04 µmol^−^, 178% higher than that of the fresh samples, which was 5.26 ± 0.20 µmol g^−1^. In contrast, the ACO samples had significantly lower levels of fagaramide (*p* < 0.01) and pellitorine (*p* < 0.001): mean pellitorine concentration of AFT1 was 43.2 ± 2.0 µmol g^−1^, or >4000% of the ACO concentration of 0.99 ± 0.00 µmol g^−1^.

We quantified the same compounds, together with yangambin, dopamine and trigonelline in the methanolic extracts (MCO, MFA, and MFP; Figure 4B). There were no significant differences in the concentrations of myristicin and elemicin amongst any of the samples. MCO samples had significantly lower levels of fagaramide, yangambin, dopamine and pellitorine than either of the fresh leaf samples, MFA and MFP (*p* < 0.001), and significantly higher levels of vitexin and trigonelline than fresh leaf samples, MFA (*p* < 0.001) and MFP (*p* < 0.01). There were no significant differences in any of the phytochemical concentrations between MFA and MPO samples.

## 4. Discussion

Our investigation of the kawakawa leaf chemical space significantly builds on the work of Obst et al. [4] and Butts et al. [5]. We have confirmed the identities of eight compounds in kawakawa and assigned identities to 58 others with confidence. We have explored the variation in commercial processed kawakawa leaf chemistry and in fresh leaves harvested over several geographic locations and seasons. Finally, we provide absolute quantification of eight of the major kawakawa leaf compounds enabling us to make inferences about the risk and benefits associated with kawakawa consumption. These results provide novel insights into the complexity and functionality of kawakawa leaf chemistry and that of the *Piper* genus.

The 64 features, including isomers, to which we have robustly assigned identities has greatly increased our understanding of the kawakawa chemical space. These features include several that have been reported in other members of the Piperaceae, but not in *P. excelsum*. Many lack common names, so we identify them here using the names of closely related structures and the differences between them. So, an analogue of ilepcimide with a saturated hydrocarbon backbone, instead of the mono-unsaturated chain, is described as “saturated ilepcimide”. The kawakawa chemical space is dominated by amides whose structures vary quite systematically. They mostly consist of a cinnamoyl, methylendioxycinnamoyl or piperoyl moiety and an amine bonded by an amide. From the synthesis of piperine, we can hypothesise that these structures likely form through the action of piperine synthase (piperoyl-CoA:piperidine piperoyl transferase), or the related enzyme, piperamide synthase [24]. These enzymes condense an organic acid thioesterified to coenzyme A with an amine. Piperine synthase favours piperic acid-CoA and piperidine as substrates. However, it can also utilise isobutylamine, yielding piperlonguminine 28 and pyrrolidine, resulting in piperyline. This enzyme is one of two closely related enzymes described in *P. nigrum* (black pepper) [24]. The second, piperamide synthase, is reported to have a broader range of substrates, catalysing the conjunction of many different organic acids, cyclic and linear to diverse amines. For example, it is involved in the synthesis of fatty acid amides reported in several Piperaceae [25,26], such as pellitorine **48**. Pellitorine, the amide of 2,4-decadienoic acid and isobutylamine, deserves special mention as it is one of the most abundant compounds in kawakawa. Pellitorine is a potent agonist of the TRPV1 receptor and is reported to induce a numbing or tingling sensation upon consumption. It is likely responsible, in part, for the spicy flavour of kawakawa. In addition to pellitorine, we also annotated features as the piperidine amide of decadienoic acid, neopellitorine B **49** and the pyrrolidine amide, sarmentine **46**. This diversity of form is common to many of the other amides we identified. We noted piperidine, pyrrolidine and isobutylamine forms of several other compounds including cinnamic acid, methylenedioxycinnamic acid, and piperic acid, as well as some longer-chain piperinoids such as piperdardine **45** and piperolein A **47**. This pattern suggests either that piperamide synthase dominates amide synthesis in *P. excelsum*, or that the synthesis of the piperic acid-CoA precursor is greatly reduced.

Cyclic amine moieties of the cinnamoyl amides often exhibited functionalisation, such as unsaturated carbon bonds, or oxidations in the form of hydroxyl or ketone groups. Notably, these functionalisations were not observed in any other amides. It may be that, despite a common synthetic pathway for all the amides, the cinnamoyl ones are further metabolised following amide formation. Or they may be products of a distinct synthetic pathway or locus.

Hydroxyls and methoxy groups were also observed on the aromatic rings of cinnamoyl moieties. This is likely to be produced by similar synthetic processes to two other major kawakawa metabolites, myristicin **44** and elemicin **37**. Interestingly these two compounds both feature triple modifications to the aromatic ring: three methoxy groups in the case of elemicin; a methylenedioxy bridge and a methoxy group for myristicin. Piperic acid is synthesised in *P. nigrum* by the enzyme CYP719A37, which forms the methylenedioxy bridge [27]. This enzyme has considerable selectivity for vanilloid precursors with an aliphatic chain length of five carbons or more. The majority of the amides we observed here exhibited three carbons in the aliphatic chain, so another variant of this enzyme is likely responsible for this transformation in *P. excelsum*. The biosynthesis of myristicin does not seem to have been elucidated, so it is not clear whether this putative enzyme is also responsible for forming the methylenedioxy bridge in myristicin. No amides with three modifications to the phenyl ring were observed in kawakawa. This, together with the abundance of unmodified cinnamoyl amides, indicates that the synthetic pathways of the alkenylbenzenes and amides diverge early in the phenylpropanoid pathways, as described in Vassao et al. [28].

Another unusual compound reported here is the anticonvulsant compound ilepcimide **25**. Ilepcimide is prescribed as a medicine to epileptics in China, hence its synonym of antiepilepsirine [29]. The compound is a truncated analogue of piperine and appears to be one of the most abundant compounds in some of the samples analysed here. The phenylpropyl amides awaine **11**, cinnamoyl piperidine **26** and cinnamoyl piperideine **16** were also in the ten most abundant features in the relative quantification data but are relatively obscure. There is little information about their pharmacology or health effects associated with their consumption in the literature. It has been reported that cinnamoylpyrrolidine **23**, a closely related compound that was also identified in our extracts at a much lower abundance than the piperidine and piperideine analogues, had greater anti-inflammatory activity than aspirin in cellular assays of platelet aggregation [30].

Several fatty amides were identified in the current work, including remarkable concentrations of pellitorine. This amide of decadienoic acid and isobutylamine has been shown to obstruct the interaction between proteins that regulate pro-inflammatory cell-adhesion [30]. It can reduce the uptake of glucose and fatty acids by human epithelial Caco-2 cells [4,31] and has been shown to reduce lipid accumulation at nanomolar concentrations [32]. Pellitorine has been shown to be a potent agonist of TRPV1, as well as the Transient Receptor Potential Ankyrin 1 receptor. Pellitorine has a numbing effect and may contribute to the analgesic effect of kawakawa that make it useful for toothache in Rongoā Māori [1].

The semi-quantitative analyses reported in our study (Table 2) improve upon those reported previously in the confidence of our compound identification and the quantity of features identified. For example, based on our metabolite search with the characteristic fragmentation ion, features with *m/z* 232.1337 reported as an amide and *m/z* 230.1181 as piperchabamide by [5] were annotated as awaine **11** and cinnamoyl piperidine **26**, respectively. Several other features that previously were identified no further than “amides” were annotated as dimethoxycinnamoyl piperidine **22** and piperanine **41** [5]. Using chromatographic deconvolution a feature with the same *m/z* and fragmentation pattern as myristicin, but considerably greater in abundance and with a different retention time, was demonstrated to be an artifact from in-source fragmentation of the lignan magnolin and not the myristicin isomer as reported earlier [5] (Appendix A). We also have confirmed the absence of apiole and piperlongumine verified by reference standard analysis.

We report the presence of the catecholamine neurotransmitter, dopamine **4**, for the first time in the Piperaceae. Dopamine has been reported in several other food plants and is suggested to function as a herbivore deterrent in algae [33]. Dopamine is present in kawakawa at concentrations of several hundred µmol g^−1^. The abundance of dopamine receptors in the gut provides a mechanism by which the consumption of kawakawa tea soothes upset stomachs and other gastrointestinal complaints in Rongoā Māori [1].

It should not be surprising that the chemical profile of commercial dried kawakawa leaves was different to that of fresh leaves. This could be due to disparity in the post-harvesting processing, genetic differences between kawakawa genotypes or other biotic and abiotic factors (or a combination thereof). It was unexpected, however, that the chemical profiles of leaves from different sources and seasons also varied as substantially as they did. The locations from which kawakawa were sampled for this analysis were selected based on traditional harvest, pragmatic and commercial considerations, but the populations are located closely enough to experience gene flow, and their ecotypes were not observed to differ radically. Herbivore damage can cause plants to raise concentrations of secondary metabolites as a defence [34,35]. However, field and laboratory experiments [36] have found no effect of prior looper caterpillar (*Cleora scriptaria*) herbivory upon the palatability of kawakawa leaves to subsequent herbivores. It has not been ascertained whether herbivory influences the flavour or pharmacology of kawakawa leaves. In the current study, we only provide preliminary information of the effects of seasonal and geographical variation on chemical profile. Having established that these factors are important, a comprehensive study that quantifies the importance of abiotic, biotic and genetic factors on chemical profile of kawakawa is warranted.

Regulatory approval in Aotearoa New Zealand for the sale of food products containing kawakawa currently is restricted to dried leaf herbal infusion (similar to tea) or dried leaf as a component of a seasoning [37]. It is important to understand the absolute concentrations of compounds within these sample types that have previously been associated with biological activity and toxicity. The concentrations of metabolites that were quantified would appear to be well below what could be speculated as toxic for humans. For example, it has been estimated that 31–36 µmol kg^−1^ day^−1^ of myristicin **44** could cause psychopharmacological effects in humans [38]. This equates to a dose of 2170–2520 µmol for a 70 kg adult. In rats, elemicin **37** was observed to induce toxicity through metabolic bioactivation upon administration of 2404 µmol kg^−1^ of elemicin [39]. Using the conversion factor from rats to humans supplied by [40], this equates to >27,000 µmol for a 70 kg human adult. Based upon the data provided above, 1 g of the kawakawa leaves infused in a 250 mL volume of hot water will release up to 5.5 μmol of myristicin and 9.0 μmol of elemicin. A gram of dried leaf powder as seasoning would contain up to 8.0 µmol elemicin and 7.1 µmol myristicin. These values are all <1% of concentrations associated with toxicity.

Comparisons with the real-world consumption of food containing elemicin and myristicin, such as carrots, are also reassuring [41]. Adult carrot consumption in Aotearoa New Zealand is estimated at 14.5 g d^−1^ [42]. Using literature values for the elemicin and myristicin content of carrots, we can estimate daily intakes of elemicin at 68 µmol [42] and myristicin at 7 µmol [43]. These amounts are greater than, or similar to intake from our values for kawakawa, so its consumption is very unlikely to be hazardous.

A no observable adverse effect level [NOAEL] of 22.3 µmol kg^−1^ day^−1^ of pellitorine **48**, or 1561 μmol for a 70 kg person, was reported from an in vivo animal study [44]. Kawakawa tea would provide 173 μmol and a gram of kawakawa leaf powder up to 43 µmol. What data exist on the risks associated with consuming plant material containing vitexin **7** and yangambin **32** also suggests that kawakawa presents little risk [45,46]. The pharmacology of ilepcimide **25** is not well documented, but experimental doses of 38 µmol kg^−1^ day^−1^ have been prescribed to children with no reported adverse effects [47]. As we could not acquire an authentic standard for ilepcimide, we used fagaramide **24** as a proxy for ilepcimide quantification. Fagaramide is an almost identical compound, differing in its isobutylamine moiety instead of ilepcimide’s piperidine ring (Appendix A). Under electrospray ionisation, they will have similar response factors, the four isomers elute within 33 s of each other, and they appear in similar relative abundance. We estimate the concentration of ilepcimide isomers in kawakawa to be 5.7 µmol g^−1^. This is less than 1% of the dose for a 20 kg child, so is unlikely to be hazardous.

There is commercial interest in incorporating kawakawa leaf powder directly into food products. It is not known how bioavailable the metabolites in such material are. Of significance, the fatty amide pellitorine does not exert the same pungency as capsaicin or piperine when consumed orally, although it does have a remarkable tingling and numbing effect [47]. This may make kawakawa more appealing as a dietary supplement to effect TRPV1 activation than capsaicin- or piperine-rich ingredients.

### Limitations

This study has used cutting-edge analytical techniques to explore the chemical space of kawakawa. However, there were limitations in the sampling of kawakawa populations, which was restricted to the Te Tauihi region. Our data do not cover the entire range of seasons. The sample sets for methanolic and aqueous extractions only marginally overlapped, limiting the comparisons that could be made between the two sample sets. The compounds that could be absolutely quantified include several of the features that exhibited the highest ion counts in the relative quantification analysis, but there are other features that may be present at even higher concentrations. These include compounds that are known to be pharmacologically active, such as ilepcimide. Future studies should focus on synthesising such compounds and quantifying them absolutely in kawakawa.

## 5. Conclusions

We report the chemical profiles of methanolic and aqueous extracts of leaves of kawakawa, a plant endemic to Aotearoa New Zealand and of great cultural importance to Māori. LC-MS/MS-based metabolic profiling combined with molecular networking and multivariate analysis reveals geographical and seasonal variation in kawakawa chemistry. With its rich phytochemical diversity, and as a candidate for functional food and ingredient applications, the further exploration of the biological properties of kawakawa could elucidate its health effects, including the mechanism underlying its beneficial effects upon inflammation and glycaemic regulation. Phytochemical concentrations in kawakawa are unlikely to be hazardous for human consumers when consumed as an aqueous infusion or as a dry leaf powder.

## Figures and Tables

**Figure 1 nutrients-14-05168-f001:**
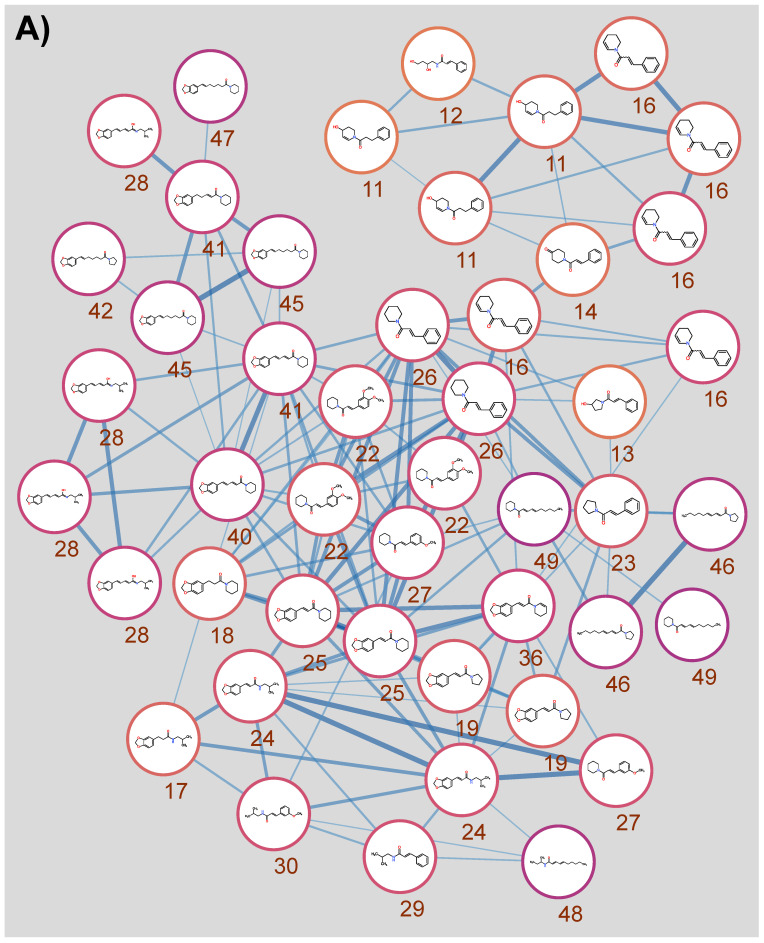
Molecular networking of LC-MS/MS features in kawakawa extracts. Nodes are unique features resolved by retention time or *m/z*. Numbers correspond to Table 2. Node borders are coloured by retention time from yellow to purple. Edge width indicates dot product 0.75–1. (**A**) The main cluster contains features that were either identified as or annotated as amides. (**B**) Features not connected to the main network of amides. Minor clusters include flavonoids, lignans and several amides. Six nodes were orphans, including several amides not connected to the main network, a lignan, phenylpropanoids and the unique compound trigonelline.

**Figure 2 nutrients-14-05168-f002:**
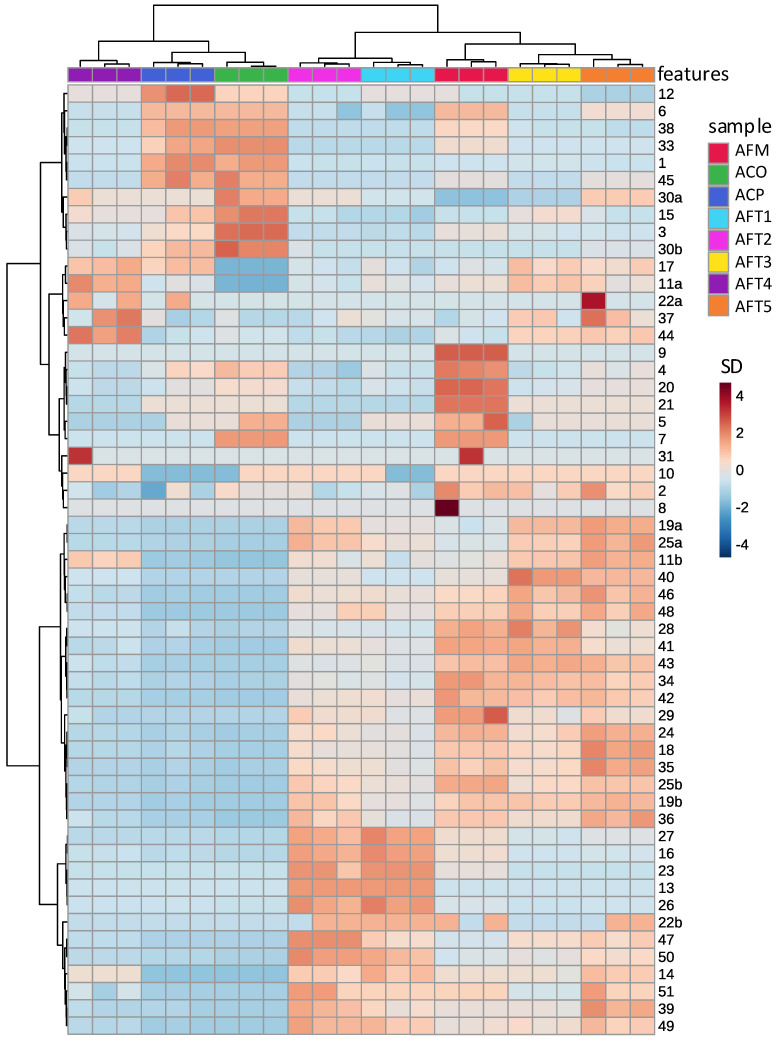
Clustergram of normalised and autoscaled peak areas from LC-MS/MS analysis of aqueous extracts. Each row represents a resolved feature (Table 2) and each column represents one triplicate analysis from the eight different leaf sources. Heat scale units are standard deviations from the mean of all peaks for that feature. Hierarchical clustering on each axis demonstrates similarity of features (*y* axis) and samples (*x* axis). The first letter of the sample label indicates aqueous extraction (A), the second letter commercial dried leaves (C) or fresh leaves (F) and the third the source as listed in Table 1. Te Tauihu samples were collected on multiple occasions throughout 2021, so a number is added to index this.

**Figure 3 nutrients-14-05168-f003:**
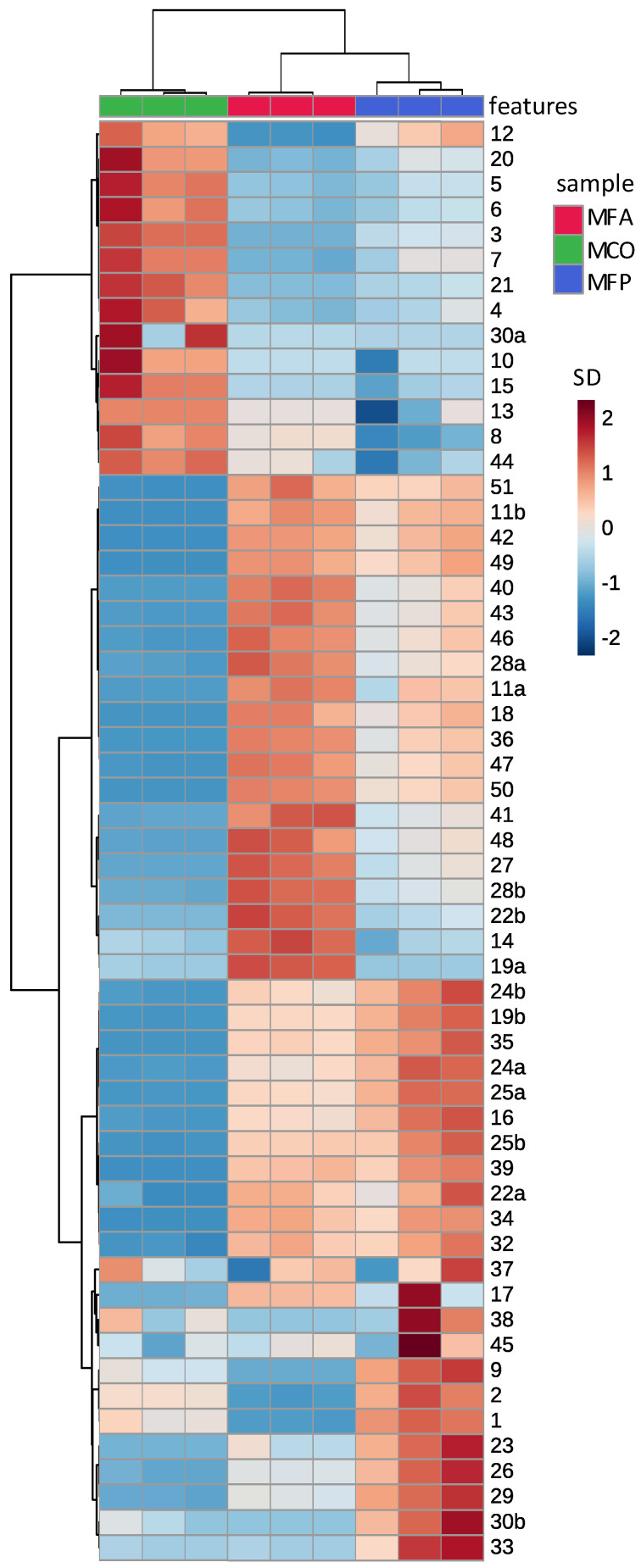
Clustergram derived from the area ratio of the phytochemicals estimated from the LC-MS/MS analysis of triplicate methanolic extracts of three types of kawakawa. Each row represents a resolved feature (Table 2) and each column represents one triplicate analysis from the eight different leaf sources. Heat scale units are standard deviations from the mean of all peaks for that feature. Hierarchical clustering on each axis demonstrates similarity of features (*y* axis) and samples (*x* axis). The first letter of the sample label indicates methanolic, the second letter commercial dried leaves (C) or fresh leaves (F) and the third the source as listed in Table 1.

**Figure 4 nutrients-14-05168-f004:**
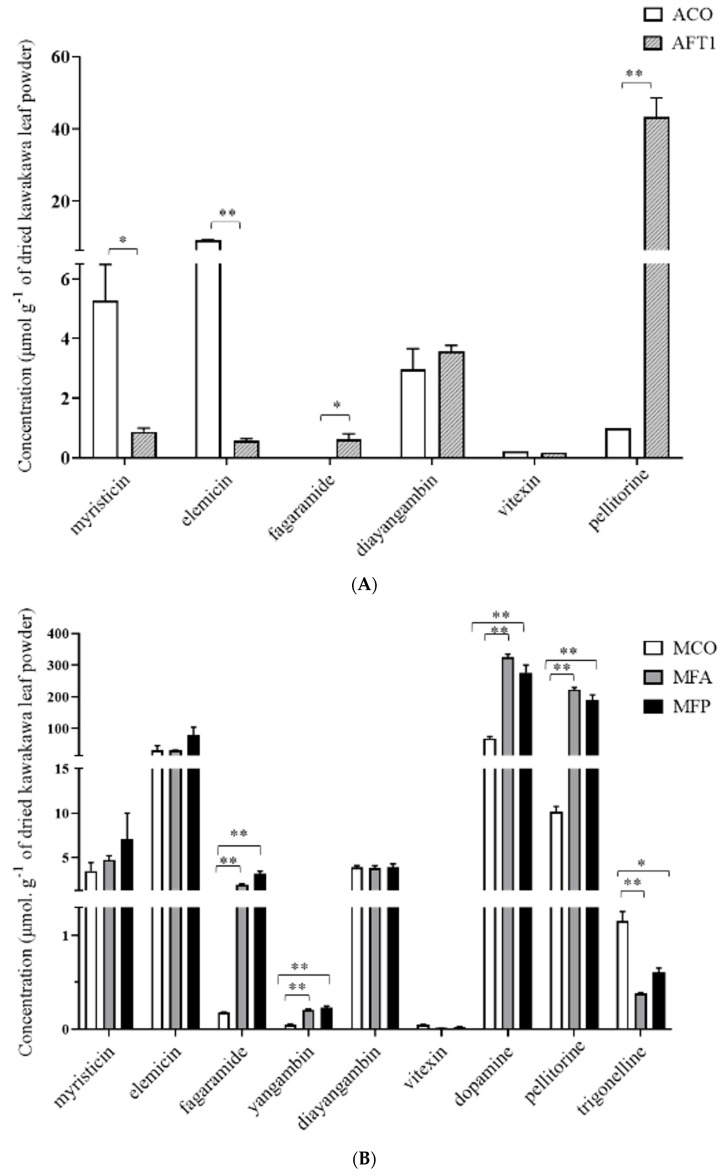
Absolute quantification of kawakawa phytochemicals obtained from (**A**) aqueous extraction of ŌKU leaf powder (ACO) and fresh leaves from the Nelson region (AFT1), and (**B**) methanol extraction of ŌKU leaf powder (MCO) and fresh leaf powders collected from Arapaoa Island (MFA) and Pōhara (MFP). Values are mean ± SD (*n* = 3). * indicates *p* < 0.01, ** indicates *p* < 0.001.

**Table 1 nutrients-14-05168-t001:** Labels for the different kawakawa leaf powder samples extracted and analysed here and used in the rest of the manuscript. The first letter indicates methanolic or aqueous extraction, the second letter commercial dried leaves or fresh leaves and the third the source. Te Tauihu samples were collected on multiple occasions throughout 2021, so a number is added to the label to index this.

Label	Date Collected/Acquired	Kawakawa Source	Extraction	Provenance
MCO	2020	ŌKU	80% methanol	commercial
MFA	Late 2021	Arapaoa Island	80% methanol	fresh
MFP	Late 2021	Pōhara	80% methanol	fresh
ACO	2020	ŌKU	aqueous	commercial
ACP	2020	Phytomed	aqueous	commercial
AFT1	Early January 2021	Te Tauihu	aqueous	fresh
AFT2	Late January 2021	Te Tauihu	aqueous	fresh
AFT3	Early March 2021	Te Tauihu	aqueous	fresh
AFT4	Late March 2021	Te Tauihu	aqueous	fresh
AFT5	April 2021	Te Tauihu	aqueous	fresh
AFM	October 2020	Mohua, Golden Bay	aqueous	fresh

**Table 2 nutrients-14-05168-t002:** Details of the features in the LC-MS/MS data annotated in this work. Multiple RTs indicate chromatographically resolved features with similar mass spectra assumed to be isomers. Multiple RT for a feature indicates isomers. The names of features identified to Level 1 of the Metabolomics Standards Initiative are boldened.

ID	Annotations [Empirical Observations]	Chemical Formula	RT (min)	Observed *m/z*	Mass Accuracy (ppm)	Major MS/MS Ions
1	salsolinol	C_10_H_13_NO_2_	3.28	180.1022	−1.28	85.0283, 145.0647, 163.0753, 180.1022
2	**trigonelline**	C_7_H_7_NO_2_	1.65	138.0550	−3.62	81.0698, 95.0854, 138.0555
3	**dopamine**	C_8_H_11_NO_2_	2.15	154.0868	0.00	91.0542, 137.0596, 154.0868
4	vitexin-O-glucoside [4]	C_27_H_30_O_15_	5.08	595.1666	0.55	271.0597, 313.0705, 337.0709, 397.0927, 415.1015, 433.1126, 595.1666
5	kaempferol malonyl neohesperidoside	C_30_H_32_O_18_	5.23	681.1676	0.56	271.0603, 313.0706, 415.1021, 433.1125, 561.1249, 681.1676
6	isorhamnetin galactoside rhamnoside	C_28_H_32_O_16_	5.28	625.1776	1.47	301.0703, 343.0812, 427.1017, 445.1124, 463.1235, 625.1776
7	**vitexin**	C_21_H_20_O_10_	5.32	433.1129	1.20	271.0606, 313.0712, 337.0712, 397.0908, 415.1024, 433.1129
8	hydroxycinnamoyl pyrrolidinol	C_13_H_15_NO_3_	5.41	234.1134	−1.27	131.0491, 234.1134
9	malonylgenistin	C_24_H_22_O_13_	5.60	519.1140	1.67	283.0595, 295.0602, 313.0702, 519.1140
10	*N*-methyldopamine	C_9_H_13_NO_2_	5.82	168.1020	0.40	91.0541, 119.0490, 151.0752, 170.0961
11	awaine [23]	C_14_H_17_NO_2_	6.62, 6.94, 7.08	232.1342	−2.32	82.0654, 105.0702, 133.0652, 150.0918, 232.1331
12	piperlotine-K	C_13_H_17_NO_3_	5.93	236.1291	2.11	86.0600, 105.0698, 133.0645, 235.1691
13	cinnamoyl hydroxypyrrolidine	C_13_H_15_NO_2_	6.13	218.1185	1.83	131.0490, 218.1174
14	cinnamoyl piperidone *(annotated as piperchabamide A)* [19]	C_14_H_15_NO_2_	6.22, 7.04	230.1184	2.12	82.0653, 100.0706, 131.0494
15	tryptophan	C_11_H_12_N_2_O_2_	6.27	205.0976	1.83	146.0598, 159.0914, 188.0704
16	cinnamoyl piperideine	C_14_H_15_NO	8.91, 9.44	214.1228	1.39	84.0807, 131.0490, 214.1223
17	saturated fagaramide	C_14_H_19_NO_3_	7.17	250.1443	−0.04	82.0651, 93.0366, 155.0853, 177.0545, 229.1034, 250.1443
18	saturated ilepcimide	C_15_H_19_NO_3_	7.36	262.1442	−0.59	82.0651, 145.0283, 177.0546, 262.1438
19	methylenedioxycinnamoyl pyrrolidine	C_14_H_15_NO_3_	7.41, 7.82	246.1130	−1.40	82.0651, 145.0282, 175.0388, 246.1130
20	orientin-O-glucoside [4]	C_27_H_30_O_16_	7.55	611.1619	−0.04	329.0663, 431.0981, 449.1084
21	kaempferol malonylglucoside glucoside	C_30_H_32_O_19_	7.64	697.1620	−0.27	127.0389, 287.0548, 329.0654, 431.0967, 554.4190
22	dimethoxycinnamoyl piperidine	C_16_H_21_NO_3_	7.74, 8.16	276.1604	−0.04	84.0807, 112.0755, 191.0700, 216.1378, 276.1599
23	cinnamoyl pyrrolidine	C_13_H_15_NO	7.89	202.1232	1.15	131.0491, 161.0959, 181.0107, 202.1225
24	**fagaramide** [4]	C_14_H_17_NO_3_	8.15 (cis), 8.40 (trans)	248.1287	0.24	175.0395, 248.1289
25	ilepcimide [4]	C_15_H_17_NO_3_	8.30, 8.70	260.1289	0.56	175.0394, 260.1288
26	cinnamoyl piperidine	C_14_H_17_NO	8.43, 8.82	216.1387	1.70	131.0494, 216.1389
27	methoxycinnamoyl piperidine	C_15_H_19_NO_2_	8.48, 8.86	246.1494	0.35	82.0650, 112.0755, 161.0596, 246.1494
28	piperlonguminine	C_16_H_19_NO_3_	8.75	274.1444	0.24	115.0534, 135.0439, 201.0544, 274.1443
29	cinnamoyl isobutylamine	C_13_H_17_NO	8.51	204.1387	−0.49	116.0529, 131.0491, 148.0755, 204.1388
30	dihydrocinnamoyl hydroxypiperidine	C_14_H_19_NO_2_	8.55, 8.76	234.1499	1.08	82.0652, 135.0802, 161.0596, 234.1499
31	dicinnamoyl putrescine	C_22_H_24_N_2_O_2_	8.75	349.1931	4.3	72.0807, 131.0490, 202.1225, 219.1492, 349.1931
32	**yangambin**, diayangambin, epiyangambin	C_24_H_30_O_8_	9.11, 9.32, 9.40	447.2023	−0.60	181.0864, 204.0787, 219.1021, 231.1022, 358.1422
33	magnolin	C_23_H_28_O_7_	9.17	417.1911	−0.12	135.0349, 151.0800, 189.0900, 193.0864, 328.13, 399.1797
34	dihydropiperlonguminine	C_16_H_21_NO_3_	9.18	276.1600	0.29	135.0444, 276.1603
35	methylenedioxycinnamoyl isopentylamine	C_15_H_19_NO_3_	9.18	262.1441	0.20	145.0283, 175.0389, 262.1438
36	ilepcimide piperideine	C_15_H_15_NO_3_	9.30	258.1129	2.26	167.0704, 175.0394, 199.01329, 257.1282, 258.1130
37	**elemicin** [3]	C_12_H_16_O_3_	9.33	209.1173	4.24	168.0785, 194.0942, 209.1178
38	N-isobutyl octadienamide	C_12_H_21_NO	9.40	196.1699	1.12	123.0804, 181.1222
39	piperamide-C7:2	C_18_H_21_NO_3_	9.40	300.1592	−0.48	74.0963, 135.0439, 302.1747
40	**piperine** [4]	C_17_H_19_NO_3_	9.59, 9.70	286.1444	0.25	201.0550, 286.1444
41	piperanine	C_17_H_21_NO_3_	9.43	288.1599	−0.76	86.0966, 135.0443, 288.1601
42	chingchengenamide A [22]	C_18_H_23_NO_3_	9.69	302.1759	−0.58	135.0444, 302.1761
43	episesartemin [4]	C_23_H_26_O_8_	9.83	431.1709	−1.96	216.1388
44	**myristicin** [4]	C_11_H_12_O_3_	10.05	193.0863	0.88	135.0808, 152.0471, 163.0757, 193.0864
45	piperdardine	C_19_H_23_NO_3_	10.45	314.1758	−0.82	135.0443, 314.1758
46	sarmentine	C_14_H_23_NO	10.63	222.1858	−2.40	98.0600, 133.1010, 151.1116, 168.1381, 222.1857
47	piperolein A	C_19_H_25_NO_3_	10.71	316.1917	0.17	86.0963, 135.0439, 173.0951, 316.1906
48	**pellitorine** [4]	C_14_H_25_NO	10.92	224.2013	0.62	209.1653, 224.2012, 227.1758
49	neopellitorine B *(annotated as achilleamide)* [4]	C_15_H_25_NO	11.37, 11.49	236.2011	0.03	123.0808, 151.1122, 236.2015
50	kalecide [4]	C_16_H_29_NO	12.33	252.2327	0.99	123.0807, 179.1071, 195.1013, 252.2328

## Data Availability

Restrictions apply to the availability of the data acquired in this research. Contact the corresponding author for more information.

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
