# Peer review of "Exploring the Chemical Space of Kawakawa Leaf (Piper excelsum)"

_nutrients, 2022, doi:10.3390/nu14235168_

Round 1

Reviewer 1 Report

The quality of the current study is high enough for acceptance:

There are some issues that need to be addressed before the manuscript is ready for publication. please find below:

- The different geographic locations in Aotearoa New Zealand, and at different times of the year affect the chemical compounds extracted from both commercially available dried kawakawa leaves, and from fresh leaves, or no? and yes, its interested for the readers.

- The introduction ought to be modified to explain the novelty point for the readers. The current manuscript has a novelty point, where the chemical Space of Kawakawa Leaf from different origins thorough the year was not considered before.

- The manuscript was well designed and written, and most importantly it's understandable for the ordinary readers; however, the language still needs to be improved.

- The MS peaks should be included herein to make sure about the current results.

Author Response

We thank the three reviewers for their comments.  We have carefully considered all comments and revised the manuscript accordingly which has significantly improved the manuscript. 

Reviewer 1

- The different geographic locations in Aotearoa New Zealand, and at different times of the year affect the chemical compounds extracted from both commercially available dried kawakawa leaves, and from fresh leaves, or no? and yes, its interested for the readers.

We discuss possible effects of biotic (especially herbivores) and abiotic (season and location) in the discussion (lines 529-540 ).   A more comprehensive analyses is required, as we comment in the text (lines 542-545  ).

- The introduction ought to be modified to explain the novelty point for the readers. The current manuscript has a novelty point, where the chemical Space of Kawakawa Leaf from different origins thorough the year was not considered before.

We include reference to geographic and seasonal variation in the introduction lines 84-86, but have also added text in the introduction to stress the novelty of study (lines 90-94 ).

- The manuscript was well designed and written, and most importantly it's understandable for the ordinary readers; however, the language still needs to be improved.

We have edited the manuscript to improve clarity

- The MS peaks should be included herein to make sure about the current results.

We do not think it is informative to readers to include ‘MS peaks’.  In our manuscript we follow accepted practice of other leading journals in listing parental m/z and fragment ions, and provide comprehensive details of interpretation of mass spec data in the materials and methods section (lines 161-207) which would enable other researchers to repeat our analyses.  Any further details of specific mass spectra can be obtained from the communicating author.

Reviewer 2 Report

Dear Authors,

Presented manuscript deals with chemical profiles of Piper excelsum leaves analyzed through targeted and non-targeted LC-MS/MS. Although the topic is very interesting, as valorisation of biocultural heritage related to human health is important both to industry and society, the manuscript needs substantial revision so to be published in a quality journal like Nutrients. Authors employed great deal of analytical methods/data analyses but failed to present the merit of their work and simply state that "Samples were collected for analysis in a somewhat ad hoc fashion, and not systematically. Our data does not cover the entire range of seasons. The sample sets for methanolic and aqueous extractions only marginally overlapped, limiting the comparisons that could be made between the two sample sets.", which is rather unacceptable.

Hopefully the Authors would improve the manuscript and I would like suggest:
1) to omit the samples extracted with methanol - obviously a downside of the experiment; to present in detail traditional consumption so to substantiate the preparation of aqueous extracts, including consumption of fresh/dry leaves;

2) to include map of the distribution and to explain the reasoning behind their interest in the sampled populations;

3) to re-structure Results and Discussion, focusing on discussing biosynthetic pathways or biological activity/safety, not both. Comparison between traditionally prepared and commercial preparations would be valuable.

Sincerely,

Author Response

We thank the three reviewers for their comments.  We have carefully considered all comments and revised the manuscript accordingly which has significantly improved the manuscript. 

Presented manuscript deals with chemical profiles of Piper excelsum leaves analyzed through targeted and non-targeted LC-MS/MS. Although the topic is very interesting, as valorisation of biocultural heritage related to human health is important both to industry and society, the manuscript needs substantial revision so to be published in a quality journal like Nutrients. Authors employed great deal of analytical methods/data analyses but failed to present the merit of their work and simply state that "Samples were collected for analysis in a somewhat ad hoc fashion, and not systematically. Our data does not cover the entire range of seasons. The sample sets for methanolic and aqueous extractions only marginally overlapped, limiting the comparisons that could be made between the two sample sets.", which is rather unacceptable.

We thank the reviewer for her/his comments and agree with the reviewer regarding the importance of the topic.  The main aim of the research described in the manuscript is to provide a comprehensive profile of the chemical composition of Piper excelsum. We provide limited information regarding the effect of seasonal and geographical variation to demonstrate the importance of these factors.  We consider that the preliminary data provided in our paper on seasonal and geographical variation alongside the detailed chemical profiling provides the basis to undertake a thorough and comprehensive study of biotic and abiotic factors which was outside the scope of the current research. We have edited the text accordingly (line  542-545 and lines 593-594)

Hopefully the Authors would improve the manuscript and I would like suggest:
1) to omit the samples extracted with methanol - obviously a downside of the experiment; to present in detail traditional consumption so to substantiate the preparation of aqueous extracts, including consumption of fresh/dry leaves;

The focus has been on aqueous extracts, as this relates to the manner by which P.excelsum is consumed, and is most relevant to our on-going human intervention studies. However, P.excelsum is also used in other forms traditional and non-traditional therapeutic preparations including those with organic solvents.  Hence, we do include a comparison between aqueous and methanol extracts, and show that certain compounds such as the alkylamide pellitorin which is an important component of P.excelsum leaf chemistry is poorly extracted in hot water.  We therefore seek to retain data from both the methanolic and aqueous extraction in the manuscript as it is informative to the readership.  There would be nothing to be gained by excluding data from the methanolic extractions.

2) to include map of the distribution and to explain the reasoning behind their interest in the sampled populations;

We have included latitude and longitude coordinates within the text (lines 115-127) as opposed to a map. Sampling of kawakawa requires permission of Iwi from the specific region.  Within this initial study, samples were only collected from the Te Tauihu region were our collaborators had the required permission to sample plants. Text has been edited accordingly (593-594).

3) to re-structure Results and Discussion, focusing on discussing biosynthetic pathways or biological activity/safety, not both. Comparison between traditionally prepared and commercial preparations would be valuable.

The aim of this paper is to provide a comprehensive chemical analyses of leaves of P.excelsum. We consider it is important to provide a discussion of biosynthetic pathways and biological activity/ safety issues. It would be significantly remiss to remove one of these areas of discussion - nothing would be gained but much information would be lost.  We agree with the reviewer that a comparison of traditional and commercial preparation would be of value. However, this is beyond the scope of the current work.

Reviewer 3 Report

This is an interesting article about a local herb in New Zealand.

Line numbers should be added.

'2.4.1. Aqueous Extraction: All leaf samples were finely ground in liquid nitrogen.' Were the leaves dried before liquid nitrogen grounding?

'Table 1' methanol should be 80% methanol.

'2.5. LC-MS/MS instrument and conditions'. Where did the ms fragments databases come from?

Page 8, for compounds from  Table 2, the authors need to discuss more the compounds identified in this study by comparing them with published papers, then draw the novelty of the study.

Page 13. 'The exceptions were myristicin and elemicin, which were slightly elevated in aqueous extracts.' What could be the reason? Do they have higher water solubility? Please cite references and discuss more them.

Page 16. 'A mass of 1 g of the kawakawa leaves studied here infused in a 250 mL volume of hot water will release up to 5.5 mol of myristicin and 9.0 mol of elemicin.' Does this seem to need a reference?

Need more recently 3-5 years published references.

Author Response

We thank the three reviewers for their comments.  We have carefully considered all comments and revised the manuscript accordingly which has significantly improved the manuscript. 

Reviewer 3

line numbers should be added.

done

'2.4.1. Aqueous Extraction: All leaf samples were finely ground in liquid nitrogen.' Were the leaves dried before liquid nitrogen grounding?

All tissue was dried before grinding (line 128)

'Table 1' methanol should be 80% methanol.

Table 1 has been edited

'2.5. LC-MS/MS instrument and conditions'. Where did the ms fragments databases come from?

-The source of the MS/MS library used for this analysis is specified in Sections 2.2 and 2.5. It was an in-house library based on the 2020 NIST MS/MS library supplemented with spectra from several authentic standards purchased specifically for this analysis. Further detail regarding the chemoinformatic tools used to annotate the features we report can be found in Section 3.1

Page 8, for compounds from Table 2, the authors need to discuss more the compounds identified in this study by comparing them with published papers, then draw the novelty of the study.

Table 2 has been edited to include reference if the compound had previously been identified in Piper. In the introduction and discussion we clearly state that we have identified 58 compounds with confidence that had not previously been identified in Piper excelsum (lines 90-94 and lines 427-434) to emphasise the novelty of the study.

Page 13. 'The exceptions were myristicin and elemicin, which were slightly elevated in aqueous extracts.' What could be the reason? Do they have higher water solubility? Please cite references and discuss more them.

Myristicin and elemicin can be considered to be compound of ‘mid polarity;, so it is not surprising that they were slightly elevated in the aqueous extracts compared to methanol extracts. Text has been edited (lines 394-395)

Page 16. 'A mass of 1 g of the kawakawa leaves studied here infused in a 250 mL volume of hot water will release up to 5.5 mmol of myristicin and 9.0 mmol of elemicin.' Does this seem to need a reference?

This is based upon the data provide in the manuscript. Text has been edited for clarity (line 557)

Need more recently 3-5 years published references.

We consider we have provided comprehensive and relevant references and further references are not required.

Round 2

Reviewer 2 Report

--